# Lower Carbon, Stronger Nation: Exploring Sociopolitical Determinants for the Chinese Public’s Climate Attitudes

**DOI:** 10.3390/ijerph20010057

**Published:** 2022-12-21

**Authors:** Yeheng Pan, Yu Xie, Hepeng Jia, Xi Luo, Ruifen Zhang

**Affiliations:** 1School of Communication, Soochow University, Suzhou 215127, China; 2Center for Science Communication & Scientific Innovation, Yangtze Delta Region Institute of Tsinghua University, Jiaxing 314100, China

**Keywords:** climate change, public attitudes, nationalism, Dual Carbon Goals, China

## Abstract

Although numerous studies have examined the Chinese public’s attitudes towards climate change, few have shed light on how sociopolitical factors related to the policy and the state have shaped such attitudes. This constituted our research goal. Against the background of China’s Dual Carbon Goals, a national survey (*n* = 1469) was conducted to investigate the relationships between climate attitudes and climate benefit perception, institutional trust, policy familiarity, nationalism, and environmental values. Findings showed that respondents shared a high score of nationalism and a high level of trust in Chinese institutions. Their national benefit perception of climate action, nationalism, and trust in national institutions were strong determinants of their attitudes towards climate change. The findings suggested that for many Chinese, a lower-carbon future will be accompanied by a coming stronger nation, which is a key driver for people to adopt positive attitudes towards addressing climate change. As such, the current study revealed an alternative landscape of the determinants underlying people’s attitudes towards climate change. To our knowledge, this is the first scholarly effort in China to empirically demonstrate the predictive role of nationalistic value in shaping climate attitudes and is one of the earliest efforts in climate communication to test the impact of the policy on such attitudes.

## 1. Introduction

In September 2020, China vowed to reach carbon emission peak by 2030 and realize carbon neutralization by 2060. Since then, the Dual Carbon Goals have quickly presided over the policy and media agenda of the world’s largest emitter [1]. Public support is crucial for China to fulfill the goals. Although numerous studies have examined the Chinese public’s attitudes towards climate change and their intention for climate-friendly behaviors [2,3,4,5], few have specifically investigated how government policies may have shaped the public attitudes and intentions, although previous studies have found that government policy can strongly shape public opinion in China [6].

Chinese people often reported high recognition of climate change and extremely high support for the government’s climate policies [7,8]. Apparently, this should be a result of the climate change narratives in the Chinese media—both in official media outlets and on social media platforms—which tend to highlight national achievements and the brilliant low-carbon future for the nation [3,9,10,11].

This led us to assume that the Chinese public’s perceived benefits of climate actions should influence their attitude towards climate change. In the collectivistic culture and with the repeated media narratives of the national achievements in addressing climate change, it is no surprise that the public should primarily consider national or collective gains as the benefits of fighting climate change. In addition to the perceived benefits, familiarity with policies is also important. Previous studies have found policy familiarity breeds trust and issue salience, which were associated with policy support [12,13]. In the climate change issue, a well-informed public was more likely to agree with the decisions of scientists and policy experts [14]. A primary goal of this study is to investigate how government policy shapes people’s attitudes towards climate change. Therefore, we put familiarity with the Dual Carbon Goals into the scope of our investigation.

Chinese media’s highlighting of the achievements of China in addressing climate change is full of nationalistic tones [9]. Indeed, recent studies have found that nationalism can predict Chinese people’s prevention and vaccinations against COVID-19 [15,16,17,18]. Although in Europe, nationalism ideology led to climate skepticism and opposition to carbon taxes [19], the prospect for a prosperous low-carbon future often made Chinese nationalists welcome policies to address climate change [20]. On the other hand, the nationalistically styled conspiracy theory that climate change is a Western plot to curb China’s development may discourage the Chinese public from endorsing actions against climate change [21]. Therefore, it is necessary for an empirical investigation into the role of nationalism in influencing the Chinese public’s climate attitudes to take place.

While nationalism represents a political value, it is also necessary to examine how people’s cultural values may predict their attitude towards climate change. Studies have identified environmental values as the main determinant of people’s climate change beliefs across countries [22,23]. Environmental values refer to beliefs about the biosphere and the effects of human activities on the biosphere [22]. It has been found that people’s egalitarian and communitarian views were associated with their support for climate policy in China [24]. Further, the levels of Chinese climate change concern were significantly influenced by personal post-materialist values [25]. Hence, we plan to probe the role of environmental value in influencing their attitude toward climate change, too.

Previous research has linked public support for climate policies to trust in governments and a group of institutional actors such as scientists and environmental non-government organizations [26,27]. On the other hand, the lack of institutional trust explains the populistic skepticism around climate change [28,29]. Climate change is a complex issue involving multiple actors ranging from national governments to international organizations, research institutes and scientists, and civil society organizations. Therefore, it is necessary to investigate how trust in different institutional actors may influence people’s attitudes toward climate change in China.

Based on the above theoretical reasoning, this study aims at investigating the relationship between the Chinese public’s climate attitudes and sociopolitical factors related to the policy and the state. Correspondingly, our general research questions are whether climate benefit perception, institutional trust, policy familiarity, nationalism, and environmental values can predict the climate attitudes of the Chinese public.

The following Section Section 2 will introduce how we conducted the survey, measured each variable, and analyzed the data. Next, in the Section Section 3, we will present the result of descriptive analyses followed by inferential analyses. We will then discuss the findings before the work is concluded with theoretical contributions, strengths, and limitations.

## 2. Materials and Methods

### 2.1. Study Design and Measures

We hired a Shanghai-based survey firm, Diaoyanba, to carry out a nationwide online survey between 25 April and 17 May 2022. The sample’s location, age cohorts, and gender distribution matched the population demographics in the China Statistical Yearbook 2020. The school administration at a large research university in East China, to which researchers of this project were affiliated, approved the research plan to offset the lack of an institutional review board (IRB) for social sciences. Most Chinese universities do not have an IRB for social science research. In the questionnaire, we stressed anonymity and privacy protection and allowed participants to exit at any time they felt uncomfortable. We eventually gathered 1469 valid respondents.

This study aimed to assess the following six points: (1) the Chinese public’s attitude towards climate change; (2) their demographic information; (3) their perception of the benefit of resolving climate change; (4) their institutional trust and the familiarity with the “Dual Carbon Goals” policy; (5) their nationalist tendency and environmental value; (6) the influence of benefit perception for addressing climate change, policy familiarity, nationalism, environmental values, and institutional trusts on climate attitudes.

There are many mature measurements of climate change attitudes. For attitudes towards climate change, we applied measurements inspired by previous studies [30,31], asking participants on a seven-point scale (1 = complete disagreement to 7 = complete agreement) the extent to which they agree to six climate-related statements, e.g., “there is already sufficient scientific evidence to prove the existence of climate change, and human beings must take immediate and urgent action.” Cronbach’s alpha of these six questions is 0.815.

For climate benefit perception, we designed a measurement on a seven-point scale (1 = totally disagree to 7 = totally agree) consisting of three statements: “actively addressing climate change can enable China to achieve innovative development;” “actively addressing climate change can boost employment;” “aggressive action on climate change will cost many people’s jobs.” Cronbach’s alpha of these three statements is 0.661.

Regarding institutional trust, factor analysis resulted in two types of trust: Type 1, trust in the Chinese government and scientists, and Type 2, trust in foreign governments and scientists and the Intergovernmental Panel on Climate Change (IPCC). We measured the extent of trust on a seven-point scale. Guttman Split-Half Coefficient of Type 1 is 0.94, and Cronbach’s alpha of Type 2 is 0.818.

We used factual knowledge regarding the “Dual Carbon Goals” to measure respondents’ familiarity with the policy. We invented three statements about China’s Dual Carbon Goals. They either often appeared in public discourse in China or represented the common public misunderstanding of China’s low-carbon undertaking. They include: “after many years of ‘energy saving and emission reduction’ policy, China’s carbon dioxide emissions have been reduced” (wrong); “The ‘carbon neutrality’ in the Double Carbon Goals means that by 2060, China will no longer emit carbon dioxide” (wrong); “The ‘carbon peak’ in the Dual Carbon Goals refers to the peak of China’s carbon dioxide emissions by 2030”(correct). All three questions had three options: the statement is ‘wrong,’ ‘correct,’ or ‘I don’t know’. In the procedure of descriptive statistics, one point was assigned only for the correct answer and zero for the wrong and ‘don’t know’ answers.

For nationalism, we adopted a well-established measurement used in China studies [32]. The measurement includes three questions: “I would rather be a citizen of China than any other country”; “My home country is better than most other countries”; “Even if our government is wrong, it should be supported.” In order to better reflect the specific situation of China’s nationalism which has stressed loyalty to the state, we added statements reflecting patriotism adapted from other well-known papers [33,34] The statements read as “I feel great when I see the Chinese flag flying” and “most countries in the world do not treat their own citizens as well as China.” Cronbach’s alpha of these five statements is 0.856.

As in the previous study [35], we measured environmental values on a seven-point scale: “people worry too much about human progress and too little about the natural environment,” and “we care too much about the future of the natural environment and not enough about current human social issues, such as commodity prices and employment.” Guttman Split-Half Coefficient of the two statements is 0.871.

We report the English translation of the survey questions in the tables in the Section Section 3 below. All questions were translated literally unless the literal translation may be incomprehensible to English readers. In such cases, we slightly adjusted the translation without distorting the original meanings.

### 2.2. Statistical Analysis

We used SPSS Version 28.0 (IBM Corp, Armonk, NY, USA) to analyze the relationship between participants’ attitudes towards climate change and their benefit perception, policy familiarity, nationalism, environmental values, and institutional trusts with the Hierarchical Linear Regression function of SPSS. A *p*-value < 0.05 was considered statistically significant.

## 3. Results

### 3.1. Demographic Characteristics

The demographic breakdown of the studied group is presented in Table 1. Most of those surveyed were relatively young (18–24, 28.5% and 25–34, 26.9%), male (50.9%), had completed college degree education (40.5%) and junior college education (27.2%), and earned less than CNY 200,000 (86.8%) per year (USD 1 = CNY 7.11). The sample includes the majority of China’s administrative provinces, with 71% of participants living in cities and 29% in rural areas. The distribution of samples within each province is consistent with the characteristics of the country’s population as a whole.

### 3.2. Attitudes towards Climate Change

The survey data showed that the respondents were generally aware of climate change (Table 2). An average of 44.0% agreed or totally agreed with the statements that climate change is scientifically founded, urgent, human-induced, relevant to all, and carries serious consequences. In particular, 47.5% agreed or totally agreed that “there is enough scientific evidence that climate change exists, and urgent action must be taken immediately.” Further, 45.1% disagreed or totally disagreed that “climate change is a natural development process of the earth, and the impact of human activities is small.”

### 3.3. Perception of the Benefit of Addressing Climate Change

By measuring the benefit perception for dealing with climate change, we wanted to know the extent to which the respondents perceived the benefit of addressing the climate problem. Of the three statements, the first one stressed the benefit to the country, and the other two pertained to social benefit, i.e., employment.

As can be seen in Table 3, respondents showed a clear recognition of the benefit of addressing climate change to the country. Over 41.8% of the respondents agreed or totally agreed that “actively addressing climate change can enable China to achieve innovative development.” In contrast, the benefit of climate change actions to employment only received mild agreement.

### 3.4. Institutional Trust and Policy Knowledge

As indicated above, two types of institutional trusts were identified. Type 1 was trust in Chinese government and scientists, and Type 2 was trust in foreign governments, foreign scientists, and the IPCC. According to Table 4, there was a stark difference in respondents’ rates between the two types of institutional trust. Chinese government and scientists won high trust among respondents in addressing climate change (M = 5.70, SD = 1.37): 67.5% trusted or totally trusted the Chinese government, and 65.5% trusted or totally trusted Chinese scientists.

As for foreign institutions, the greatest share of respondents chose to be neutral, even though more people chose to trust rather than distrust (M = 4.45, SD = 1.21). It is noted that to respondents, the IPCC was more trustable than foreign governments and scientists.

Table 5 shows the result of participants’ policy familiarity indicated by their relevant knowledge. In general, respondents’ knowledge of China’s climate policy was quite limited. For example, for the statement “after many years of ‘energy saving and emission reduction’ policy, China’s carbon dioxide emissions have been reduced,” 65.1% could not judge whether it was correct or wrong, and only 7% successfully recognized that it was wrong. Since we assigned 1 for the correct answer and 0 for the wrong and ‘don’t know’ answers, the aggregate of three scores reflected the level of participants’ knowledge about the Dual Carbon Goals, as shown in Figure 1. The figure tells us that only 1.9% of the participants answered all three questions correctly, while 32.1% got them all wrong.

### 3.5. Nationalism and Environmental Value

Participants scored quite high on nationalism indicators (M = 5.46, SD = 1.25, See Table 6). More than 55.7% of them totally agreed with “I would rather be a citizen of China than any other country,” and 54.5% totally agreed with the statement that China is better than most other countries.

Participants also had a high endorsement of environmental value (M = 4.71, SD = 1.38, See Table 7). Over 33.6% of the respondents agreed or totally agreed with the argument that people worry too little about the natural environment.

### 3.6. The Association between Climate Benefit Perception, Institutional Trust, Policy Knowledge, Nationalism, Environmental Values, and Climate Attitudes

We used the hierarchical regression model to test the correlation between demographic factors (gender, age, education, and income), benefit perception, knowledge (about the Dual Carbon Goals), nationalism score, institutional trust, and public attitudes towards climate change. Hierarchical regression can examine how interested factors explain statistically significant variance in the dependent variable in models gradually adding more relevant variables, so that we can observe how the explanatory power of interested factors changes [36].

The results of the regression are reported in Table 8. In the first model of demographic factors, we found that gender (male = 0, *β* = 0.09, *p* < 0.001) and education (*β* = 0.16, *p* < 0.001) could affect the participants’ attitudes towards climate change. Participants with higher education backgrounds are more likely to have positive attitudes towards climate change. The demographic factors accounted for 2.3% of the variation in attitudes towards climate change.

Step 2 added benefit perception to the model. The perception was positively associated with participants’ attitudes towards climate change (*β* = 0.44, *p* < 0.001), but explaining only an additional 0.9% of the variation in attitudes.

In step 3, we added participants’ knowledge about the Dual Carbon Goals into the model, and it showed a positive association with the attitudes (*β* = 0.15, *p* < 0.001), which enhanced 18.7% of the variation.

In steps 4 and 5, we found nationalism (*β* = 0.29, *p* < 0.001) and environmental value (*β* = 0.09, *p* < 0.001) were positively associated with climate attitudes. Models 4 and 5 accounted for an additional 2.3% and 9.5% of the variation, respectively.

In step 6, we added two types of institutional trusts, and found that Type 1 (trust in Chinese government/scientists) was significantly associated with climate attitudes (*β* = 0.35, *p* < 0.001), while Type 2 (trust in foreign governments/scientists, and IPCC) was nonsignificant. Institutional trust enhanced the model’s explanatory power by 7.3%. This final model explained 40.7% of the variance in attitudes towards climate change.

## 4. Discussion and Conclusions

The findings confirm previous research discoveries that Chinese people generally had a higher recognition of climate change and high support for the country’s climate policies [7,8]. Our participants seemed to have a firm perception that the Chinese economy and society will benefit from addressing climate change. They shared a high score of nationalism and a high level of trust in Chinese institutions as in numerous other studies [15,16,17,37,38]. More importantly, respondents’ national benefit perception, nationalism, and trust in national institutions were strong determinants of their attitudes towards climate change. Put together, these factors have jointly depicted a picture in the Chinese public mind: “China has successfully marched toward a low-carbon development goal, and this development route will make the country stronger.” In a nutshell, for many Chinese, a lower-carbon future will be accompanied by a coming stronger nation.

The stronger nation imagination was embodied in the relatively high coefficients of the benefit perception for climate actions across various models. However, the mere addition of this variable can only account for a small percentage of explained variances. The explanatory power of the whole model was dramatically improved after the knowledge about the Dual Carbon Goals was brought in to assess policy familiarity in model 3. This fact indicates that given the majority of Chinese people’s long-time obedience to the government’s calls, the positive climate attitudes of many may simply be a reflection of their attitudinal orientation to be consistent with the government. The bigger momentum for positive climate attitudes depends on familiarity with and corresponding recognition of the “Dual Carbon Goals”.

Adding value factors—nationalism and environmental value—in models 4 and 5 significantly increased their explanatory power. As discussed in the Introduction, value orientation plays an important role in shaping people’s environmental attitudes. Our study is not an exception. However, different from what has been revealed by previous studies using both Chinese and Western samples [23,24,39], our study empirically highlighted the role of nationalistic value orientation for the first time. The fact that the final model, which contained several variables pointing to the nationalistic values, can explain 40.7% of the variance in attitudes towards climate change indicates a strong orientation towards nationalistic value.

While nationalism as a whole was most strongly related to attitude, environmental values also played a major role. There was not a contradiction. The two types of values can go hand in hand. To the general Chinese public, a stronger nation needs to cherish the environment, and it is this stronger nation that can respect people’s environmental values.

Although the nationalistic value and stronger nation imagination drove people to adopt a more positive attitude towards climate actions, the dramatic decline in the coefficient value of the nationalism variable in the final model, which added institutional trust, needs to be further explained. Apparently, the trust in the Chinese government and scientists accounted for the reduced coefficient value of nationalism. The trust also lowered the values of coefficients of all other variables and deprived the statistical significance of predictions by demographic factors. The reason for this is simple. Either nationalistic feelings or a stronger nation imagination need someone to realize them. The belief that this someone, here the Chinese government and scientists, can fulfill these goals was the more fundamental determinant to drive climate attitudes. Understandably, although climate change imposes global challenges and calls for the world’s joint effort, in the Chinese mindset, trust in foreign governments and even the IPCC was not the driver behind climate change attitudes.

Combining all the determinants, we can conclude that this study has fulfilled the task of examining not only individual determinants of Chinese people’s attitude towards climate change but also the overall impact of government policies. The “Dual Carbon Goals” brought in a strong nation imagination through the low-carbon route, ignited public pride in the national achievements, and rendered a trustable state to implement the goals. Although whether the nationalistic value orientation can become a consistent driver for people to struggle against climate change can be further discussed, our study has already achieved a milestone by revealing this alternative attitude-driving pattern in the climate change issue.

### Theoretical Contributions, Strengths, and Limitations

The primary theoretical contribution of the current study is to reveal an alternative landscape of the determinants underlying people’s attitudes towards climate change. Rather than relying on individualistic factors such as risk and personal benefit perceptions, this study uncovered that nationalistic value orientation can become a driving engine for people to adopt positive attitudes towards addressing climate change, at least in China. To our knowledge, this is the first scholarly effort worldwide to empirically demonstrate the predictive role of nationalistic value in shaping climate attitudes, yet it is highly plausible, given that Chinese people’s nationalistic feelings can also predict their preventive behaviors against the COVID-19 pandemic [16,17].

The second theoretical contribution of our study is to exhibit that the impact of the policy per se can influence the public attitudes targeted by the policy. Nearly all previous studies in the field of climate change public opinion used policy support for climate policies as a dependent variable to be determined by a group of individual and collective factors. However, in the circumstances of China, where policy obedience is encouraged, cognitive factors linked to the policy—self-identification with the policy goals, familiarity with policy texts, the knowledge of policy facts, and value inclination—can become determinants for people to form favorable attitudes towards the policy.

The third contribution of this research is to initially throw light upon a long-time contradiction in the field of climate communication in China. While people’s positive attitudes towards climate change and corresponding policy support in many other countries are associated with stronger perceived risk of climate [40,41], a PEW study found that the Chinese and US publics were least concerned about climate change’s impact [29]. The trend identified by PEW seemed constant, as more recent research found that Chinese respondents’ perceptions towards climate change were laxer than those in other countries [42], and their concerns for the environment in general and climate change specifically were relatively low [25,43]. Now, our data can offer an explanation for this contradiction. Chinese people’s positive attitude towards climate action is not because they personally feel the climate change risks. Rather, they support the government’s climate policies simply because these policies raise a stronger imagination of China and because their nationalistic value shaped their adherence to the state policies.

For practical implications, our study demonstrates that politicized factors triumphed over individual risk and benefit perception to dominate the Chinese public’s attitude towards climate change. However, this does not mean that personal risk and benefit perceptions should be overlooked during the campaign to mobilize Chinese people to act against global warming. Instead, we should strengthen the information to help individuals to make rational decisions. However, such an edification process can be carried out in a nationalistic tone to improve the persuasion effect in China. With more studies exploring the politicized aspect of Chinese people’s environmental behaviors, indoctrinating solid science and environment information through nationalistic or collectivistic messages might be adoptable in other environmental and health scenarios.

Despite its finding on the China-specific contributors to people’s attitudes towards climate change, this study is not without limitations. First, this study was performed in the Chinese context. Therefore, the main conclusion of our research, the decisive role of nationalism and trust in domestic institutions in influencing people’s attitudes towards climate change, may hardly be generalized. The uniqueness of the Chinese sociopolitical context and the COVID-19 setting in which our survey was conducted may have pushed higher Chinese nationalism. Therefore, we would not recommend any attempt to generalize our findings. Instead, our study further confirms that the factors influencing public attitudes towards climate change were embedded in different socioecological contexts.

Second, the survey’s cross-sectional feature restricts us from making any causal conclusions. Based on the data available to us, we cannot wholly exclude possible confounding factors that drove both the attitudes towards climate change and the viewpoints on government policies. Though it is reasonable to assume that nationalistic feeling should represent a sustaining value that can decide both attitudes, well-designed future studies are needed to provide evidence for this.

Our last limitation is the timing of the survey. The month of April when the data were collected is one of the best months of the year in terms of weather in most parts of China. This made people less aware of the impact of climate change and the natural disasters associated with it. Follow-up research should be conducted to overcome this insufficiency. These follow-up studies, with a more delicate design, broader context, and more diversified research methods, might help to address the limitations of the context and cross-sectional nature of the current survey-based research.

## Figures and Tables

**Figure 1 ijerph-20-00057-f001:**
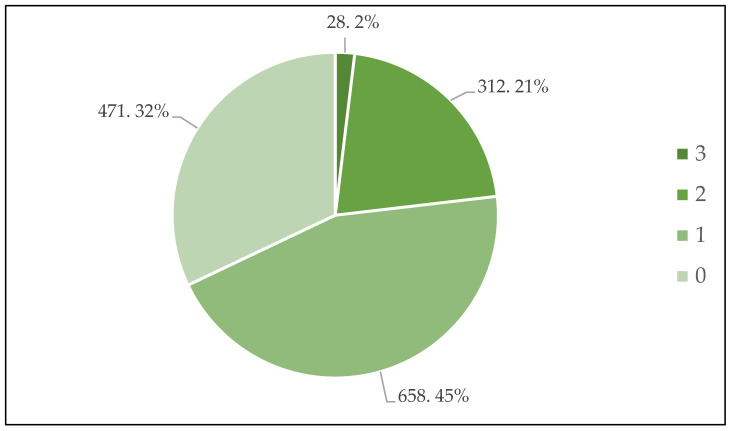
Levels of participants’ knowledge about the Dual Carbon Goals. Notes: The numbers on the pie chart read as ‘number of respondents, percentage’. The list of numbers on the right side is the aggregate of scores respondents received while answering the three questions.

**Table 1 ijerph-20-00057-t001:** Distribution of demographic characteristics of the sample (*n* = 1469).

Variable	% (*n*)
Gender	
Male	50.9 (747)
Female	49.1 (722)
Age	
<18	4.7 (69)
18–24	28.5 (418)
25–34	26.9 (395)
35–44	25.7 (378)
45–54	13.1 (193)
55–64	0.6 (9)
>64	0.5 (7)
Education Level	
Junior high school and below	9.9 (146)
Senior high school	19.1 (280)
Junior college education	27.2 (399)
College degree	40.5 (595)
Postgraduate degree	2.7 (40)
Doctoral Degree	0.6 (9)
Yearly Income (CNY)	
100,000 or less	59.2 (870)
100,001–200,000	27.6 (405)
200,001–500,000	9.6 (141)
500,001–1,000,000	2.5 (36)
1,000,001–5,000,000	0.8 (12)
More than 5,000,000	0.3 (5)

**Table 2 ijerph-20-00057-t002:** Participants’ attitudes towards climate change (*n* = 1469).

Question: Please Make Your Judgment on the Following Statements on Climate Change:							
	Totally Disagree	Disagree	A Little Disagree	Neutral	A Little Agree	Agree	Totally Agree
(1) There is already sufficient scientific evidence to prove the existence of climate change, and human beings must take immediate and urgent action.	2.2 (33)	2.4 (35)	3.5 (51)	26.7 (392)	17.7 (260)	28.1 (413)	19.4 (285)
(2) Some people’s concerns about climate change are too exaggerated, and there is no need to take special action (reversed coding).	14.0 (206)	21.3 (313)	25.5 (375)	29.4 (432)	4.6 (67)	3.2 (47)	2.0 (29)
(3) Climate change is a natural development process of the earth, and the impact of human activities can be said to be very small (reversed coding).	18.5 (272)	26.6 (391)	21.0 (308)	25.2 (370)	4.2 (61)	2.8 (41)	1.8 (26)
(4) Climate change will lead to more infectious diseases.	2.2 (32)	2.8 (41)	5.4 (80)	29.5 (433)	20.7 (304)	26.9 (395)	12.5 (184)
(5) If nothing is done, climate change will flood many of our coastal cities.	1.8 (26)	1.5 (22)	4.6 (67)	29.3 (431)	22.8 (335)	24.5 (360)	15.5 (228)
(6) Responding to climate change has little to do with us ordinary people (reversed coding).	34.3 (504)	22.2 (326)	11.7 (172)	23.5 (345)	4.6 (67)	2.5 (37)	1.2 (18)
Total							
Mean	5.12						
SD	1.00						

Notes: As Statements 2, 3, and 6 are negative about climate change, we reversed their scores when calculating the mean of climate attitudes and the average percentage of positive attitudes.

**Table 3 ijerph-20-00057-t003:** Participants’ climate benefit perception (*n* = 1469).

Question: Please Make Your Judgment on the Following Statements on Climate Change:							
	Totally Disagree	Disagree	A Little Disagree	Neutral	A Little Agree	Agree	Totally Agree
(1) Actively addressing climate change can enable China to achieve innovative development.	2.5% (36)	1.4% (21)	4.3% (63)	26.1% (384)	23.8% (350)	24.0% (353)	17.8% (262)
(2) Actively addressing climate change can boost employment.	6.9% (101)	8.8% (129)	13.6% (200)	39.5% (580)	14.6% (215)	10.4% (153)	6.2% (91)
(3) Aggressive action on climate change will cost many people’s jobs (reversed coding).	4.2% (61)	7.1% (105)	18.7% (275)	38.3% (562)	15.0% (220)	10.1% (148)	6.7% (98)
Total							
Mean	4.34						
SD	1.10						

**Table 4 ijerph-20-00057-t004:** Participants’ institutional trust (*n* = 1469).

	Question: How Much Do You Trust These Actors in Climate Change?							
		Totally Distrust	Distrust	A Little Distrust	Neutral	A Little Trust	Trust	Totally Trust
Type 1 Trust	(1) Chinese government	1.6 (23)	1.4 (20)	2.0 (30)	19.6 (288)	7.9 (116)	27.1 (398)	40.4 (594)
	(2) Chinese scientists	1.6 (23)	1.4 (20)	2.4 (35)	19.9 (293)	9.3 (136)	29.0 (426)	36.5 (536)
Type 2 Trust	(3) Foreign governments	6.5 (96)	6.1 (89)	11.0 (162)	46.7 (686)	12.0 (176)	10.3 (152)	7.4 (108)
	(4) Foreign scientists	4.9 (72)	5.2 (77)	9.5 (139)	45.3 (666)	15.9 (233)	12.1 (178)	7.1 (104)
	(5) the IPCC	2.5 (36)	1.4 (21)	4.3 (63)	34.6 (508)	18.4 (271)	24.2 (355)	14.6 (215)

**Table 5 ijerph-20-00057-t005:** Participants’ knowledge about the Dual Carbon Goals (*n* = 1469).

Questions about Dual Carbon Goals			
Question: Do you Think the Following Statement is Correct?	**Wrong**	**Correct**	**I don’t know**
After many years of “energy saving and emission reduction” policy, China’s carbon dioxide emissions have been reduced.	7.0 (103)	27.8 (409)	65.1 (957)
The “carbon neutrality” in the double carbon target means that by 2060, China will no longer emit carbon dioxide.	41.3 (607)	39.1 (574)	19.6 (288)
The “carbon peak” in the Dual Carbon Goals refers to the peak of China’s carbon dioxide emissions by 2030.	14.8 (218)	40.5 (595)	44.7 (656)

**Table 6 ijerph-20-00057-t006:** Participants’ nationalist tendency (*n* = 1469).

Question: To What Extent Do You Agree with the Following Statements?							
	Totally Disagree	Disagree	A Little Disagree	Neutral	A Little Agree	Agree	Totally Agree
(1) I would rather be a citizen of China than any other country.	4.0 (59)	2.0 (30)	1.8 (27)	12.2 (179)	8.0 (117)	16.3 (239)	55.7 (818)
(2) My home country is better than most other countries.	2.4 (35)	2.0 (30)	2.1 (31)	13.2 (194)	8.9 (131)	16.9 (248)	54.5 (800)
(3) Even if our government is wrong, it should be supported.	9.6 (141)	11.7 (172)	21.0 (308)	32.3 (474)	7.1 (104)	7.8 (114)	10.6 (156)
(4) I feel great when I see the Chinese flag flying.	2.5 (36)	1.4 (21)	1.6 (24)	8.8 (130)	6.5 (95)	14.0 (206)	65.1 (957)
(5) Most countries in the world do not treat their own citizens as well as China.	2.5 (37)	1.4 (20)	5.0 (74)	25.1 (368)	10.9 (160)	15.0 (220)	40.2 (590)
Total							
Mean	5.46						
SD	1.25						

**Table 7 ijerph-20-00057-t007:** Participants’ environmental value (*n* = 1469).

Question: To What Extent Do You Agree with the Following Statements?							
	Totally Disagree	Disagree	A Little Disagree	Neutral	A Little Agree	Agree	Totally Agree
(1) People worry too much about human progress and too little about the natural environment.	3.9 (57)	3.3 (49)	8.5 (125)	22.9 (337)	27.7 (407)	17.7 (260)	15.9 (234)
(2) We care too much about the future of the natural environment and not enough about current human social issues, such as commodity prices and employment (reversed coding).	10.0 (147)	13.7 (201)	28.4 (417)	31.7 (466)	7.7 (113)	4.6 (68)	3.9 (57)
Total							
Mean	4.71						
SD	1.38						

**Table 8 ijerph-20-00057-t008:** Hierarchical regression model of factors associated with attitudes towards climate change (*n* = 1469).

Step	Variable	Public Attitude towards Climate Change					
		Model 1	Model 2	Model 3	Model 4	Model 5	Model 6
		*β*	*β*	*β*	*β*	*β*	*β*
**1**	Demographic factors						
	Gender (Male = 0)	0.09 ***	0.06 **	0.06 **	0.05 *	0.05 *	0.04
	Age	−0.01	−0.01	−0.01	−0.01	−0.01	−0.01
	Education	0.16 ***	0.09 ***	0.07 **	0.06 *	0.05 *	0.03
	Family Income	−0.05	−0.04	−0.04	−0.03	−0.03	−0.02
**2**	Benefit perception						
	Benefit perception for climate actions		0.44 ***	0.43 ***	0.34 ***	0.34 ***	0.27 ***
**3**	Policy familiarity						
	Knowledge about the Dual Carbon Goals			0.15 ***	0.16 ***	0.15 ***	0.12 ***
**4**	Nationalism						
	Nationalism score				0.29 ***	0.22 ***	0.07 **
**5**	Environmental Value						
	Environmental value score					0.17 ***	0.12 ***
**6**	Institutional trust						
	Institutional trust—Type 1						0.35 ***
	Institutional trust—Type 2						−0.01
	Model statistics						
	N	1469	1469	1469	1469	1469	1469
	Adjusted R^2^	2.3%	3.2%	21.8%	24.0%	33.5%	40.7%
	ΔR^2^	2.4%	0.9%	18.7%	2.3%	9.5%	7.3%

* *p* < 0.05. ** *p* < 0.01. *** *p* < 0.001.

## Data Availability

Materials and anonymous data are available from the authors by request.

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
