# Peer review of "Lower Carbon, Stronger Nation: Exploring Sociopolitical Determinants for the Chinese Public’s Climate Attitudes"

_ijerph, 2022, doi:10.3390/ijerph20010057_

Round 1
Reviewer 1 Report
The purpose of the research should be clearly stated in the abstract. In table 1: Yearly income - what units is it in? Figure 1 - what do the numbers 1,2,3,4 mean in the legend of the figure? Please describe them either in the drawing or in the caption of the drawing. In the bibliography, as well as in the text, you need to replace older literature items with newer ones, e.g. items: 12, 33, 34, 42. These authors have certainly published their newer publications on this topic. After taking into account the above-mentioned comments, the manuscript can be published.
Good luck
Author Response
Our point-to-point response to Reviewer 1:
1.“The purpose of the research should be clearly stated in the abstract.”
Response: Thanks for your suggestion. We stated in the abstract “how sociopolitical factors related to the policy and the state have shaped such attitudes,” which is exactly our research goal or purpose. Now we’ve added a short sentence to accentuate it. In addition, we’ve added a single paragraph at the end of the introduction section to more clearly demonstrate our purpose of research.
2.“In table 1: Yearly income - what units is it in? “
Response: Thanks for pointing this out. We only mentioned the CNY in the text, and now, we’ve put it in the table for clarification.
3.“Figure 1 - what do the numbers 1,2,3,4 mean in the legend of the figure? Please describe them either in the drawing or in the caption of the drawing.”
Response: Thanks! As we assigned 1 for the correct answer and 0 for the wrong and ‘don't know’ answers, the list of the numbers is the aggregate of scores respondents got while answering three questions. They reflected the level of participants’ knowledge about the dual carbon goals. Now we’ve explained this in the notes of Figure 1 as well as in the text.
4.“In the bibliography, as well as in the text, you need to replace older literature items with newer ones, e.g. items: 12, 33, 34, 42.”
Response: Thanks for the suggestion! Now we’ve updated our literature.
Reviewer 2 Report
Pan et al. conducted a national survey in China with a sample of 1469 people to investigate the relationships between climate attitudes and climate benefit perception, institutional trust, policy familiarity, nationalism, and environmental values of Chinese people. The authors have shown a high score of nationalism and a level of trust in Chinese institutions, which I find obvious outcomes. What is worth here is that the respondents' attitudes to climate change depend on their national benefit perception of climate action, nationalism, and trust in national institutions. I have a few questions and suggestions for the authors.
The population in china has reached 1.4 billion. A sample of 1.5 thousand is too small to represent such a vast population. Please increase the sample size by at least 10 to 50 times.
It is apparent that the data was collected nationwide. The question is whether the respondents were selected randomly based on geographical distribution, population density, or other means. Were the respondents from urban, suburban, or rural areas? What is the respondents' distribution based on this?
I highly recommend the authors not include any political opinion in the manuscript. Please rewrite the introduction section addressing the environmental values and eliminating political views. In the discussion section, the author has written a (kind of) slogan of Chinese president Mr. Jinping. Please delete that and also the associated references.
Author Response
Our point-to-point response to Reviewer 2:
1.“The population in china has reached 1.4 billion. A sample of 1.5 thousand is too small to represent such a vast population. Please increase the sample size by at least 10 to 50 times.”
Response: We understand the concern of the reviewer. Our sample indeed seemed small against the entire Chinese population. But we would argue that the main purpose of this study is looking into the relationship between climate attitudes and sociopolitical factors. For this purpose, the size of the current sample should be okay for us to make reliable statistical inferences about the relationship. We will also adjust our wording to avoid that our findings seemed to representative. It is a quota-sample, basically not a representative sample.
2. “It is apparent that the data was collected nationwide. The question is whether the respondents were selected randomly based on geographical distribution, population density, or other means. Were the respondents from urban, suburban, or rural areas? What is the respondents' distribution based on this?”
Response: Thanks for the questions. As we explained in the Materials and Methods section, the current sample’s location, age cohorts, and gender distribution matched the population demographics in the China Statistical Yearbook 2019. We didn’t select the respondents by ourselves but commissioned the survey firm Diaoyanba to select respondents randomly from its pool in accordance with the statistical yearbook. This procedure, as well as the company’s data, have been accepted by previous publications and thus we believe in its validity. We surveyed more about demographic information than what we showed, so now we added more information of interest in the results section.
3. “I highly recommend the authors not include any political opinion in the manuscript. Please rewrite the introduction section addressing the environmental values and eliminating political views.”
Response: Thank you for your suggestions. We have rewritten the text and tried to delete the political opinion as much as possible. But some of our variables, such as nationalism, may reflect exactly a politicized dimension and it is the core of our study. Therefore, we did not change our research structure and only adjusted wording and implications as much as possible (to discuss how nationalistic feelings can be used to promote environmental behaviors). We wish this is accepted by the reviewers.
4. “In the discussion section, the author has written a (kind of) slogan of Chinese president Mr. Jinping. Please delete that and also the associated references.”
Response: We mentioned Xi’s signature policy for the purpose of explaining the nationalist inclination of our respondents. But now we’ve removed the content with references as the reviewer suggested.
Round 2
Reviewer 2 Report
The authors have adequately addressed my concerns in the revised version.